# Polyamines in Microalgae: Something Borrowed, Something New

**DOI:** 10.3390/md17010001

**Published:** 2018-12-20

**Authors:** Hung-Yun Lin, Han-Jia Lin

**Affiliations:** 1Department of Bioscience and Biotechnology, National Taiwan Ocean University, Keelung 20224, Taiwan; hungyun59@gmail.com; 2Center of Excellence for the Oceans, National Taiwan Ocean University, Keelung 20224, Taiwan

**Keywords:** bioactive compounds, microalgae, polyamines, metabolic pathways, stress response

## Abstract

Microalgae of different evolutionary origins are typically found in rivers, lakes, and oceans, providing more than 45% of global primary production. They provide not only a food source for animals, but also affect microbial ecosystems through symbioses with microorganisms or secretion of some metabolites. Derived from amino acids, polyamines are present in almost all types of organisms, where they play important roles in maintaining physiological functions or against stress. Microalgae can produce a variety of distinct polyamines, and the polyamine content is important to meet the physiological needs of microalgae and may also affect other species in the environment. In addition, some polyamines produced by microalgae have medical or nanotechnological applications. Previous studies on several types of microalgae have indicated that the putative polyamine metabolic pathways may be as complicated as the genomes of these organisms, which contain genes originating from plants, animals, and even bacteria. There are also several novel polyamine synthetic routes in microalgae. Understanding the nature of polyamines in microalgae will not only improve our knowledge of microalgal physiology and ecological function, but also provide valuable information for biotechnological applications.

## 1. Introduction

Polyamines are a group of linear or branched aliphatic compounds containing multiple amino groups. Almost all organisms, from bacteria to higher plants and animals, can synthesize polyamines, which are derived from amino acids and are considered to be an essential regulatory molecule for different physiological and pathological conditions [1,2,3,4,5,6]. To date, numerous functions have been reported for cellular polyamines, including nucleic acid protection, regulation of gene expression and protein translation, modulation of signal transduction, and cell membrane stabilization [7,8,9,10].

Since polyamines play key roles in multiple cellular functions, disruption of polyamine homeostasis has pronounced effects on cells and the physiology of individual organisms. Both *in vitro* and *in vivo* experiments have indicated that fluctuations in intracellular polyamine levels are tightly linked to regulation of the cell cycle, with inhibition of polyamine synthesis being found to impede cell proliferation [11,12]. Moreover, studies on bacteria and plants have shown that polyamine anabolism is related to the ability of these organisms to tolerate different types of environmental stress [13,14,15,16,17]. Direct application of polyamine-producing bacteria or a cyanobacterial hydrolysate to plants could promote their growth [18,19,20,21], suggesting that they may have auxin-like or cytokinin-like bioactivity [18,19]. Numerous studies in animals, including moths, fruit flies, rats, and humans, have also indicated that polyamine production declines with age [22,23,24]. Supplement of polyamine in diet can mitigate age-related memory impairment [23], and even prolong the life of animals [25,26,27]. Previous studies have also supported that multi-cellular or unicellular organisms can acquire their polyamines through feeding or by direct absorption from the environment [28,29,30]. Therefore, polyamines not only affect the physiology of individual organisms but also influence entire ecosystems through the food web.

Unicellular microalgae are widely distributed in freshwater and marine ecosystems. They can provide up to 45% of global primary production, higher than the sum of all terrestrial higher plants [31,32]. Although they have a significant impact on the global ecosystem, studies on microalgal polyamines have largely been overlooked. What types of polyamine do microalgae produce? How do polyamines affect the physiological functions of microalgae? And how do polyamines produced by microalgae affect other organisms? These questions are not only interesting to experts on microalgae but also important for ecological and biochemical studies. With the recent advancement of technology, analysis of microalgal polyamines has become easier than ever before. Common polyamines are usually detected by thin layer chromatography (TLC) or high-performance liquid chromatography (HPLC) after fluorescent derivatization [33]. Currently, gas chromatography (GC)-Mass spectrometry or LC-Mass/Mass spectrometry is also frequently employed in polyamine detection, especially uncommon polyamines, for which obtaining standard samples are difficult [34,35,36]. LC-Mass/Mass is the most sensitive polyamine detection method, in which the detection limit can be as low as 0.1 ng/mL [36]. With the analysis of polyamine contents and the sequencing of various microalgal genomes in recent years, we now have a preliminary understanding of the metabolic pathways and possible physiological functions of polyamines in microalgae. This article aims to review the recent research progress and the differences in polyamine content, diversity, and metabolic pathways between microalgae and other species.

## 2. Polyamine Composition of Microalgae

Microalgae is a term usually referring to eukaryotic unicellular organisms that contain chlorophyll. Depending to the source of the chloroplasts, microalgae can be generally divided into three categories [37]. The chloroplasts of Archaeplastida (including Glaucophyta, Rhodophyta, and Chlorophyta) originated from a primary endosymbiotic event. The ancestors of Rhodophyta and Chlorophyta also evolved into chloroplasts of other microalgae such as Rhizaria (Chlorarachniophyta), Chromalveolata (Cryptophyta, Stramenopiles, Haptophyta, Apicomplexa, Chromerida, and certain Dinoflagellata), and Excavata (Eugleonophyta) through secondary endosymbiotic events [37]. Recently, the chloroplasts of the phylum Dinoflagellate were found to contain plastids derived from Chlorophyta, Cryptophyta, Stramenopiles, and Haptophyta, suggesting that a third endosymbiotic event occurred in certain microalgae [38,39].

The genomes of microalgae are relatively diverse due to the complexity of their evolutionary origin. In addition, many bacterial genes are found in microalgal genomes as a result of horizontal gene transfer, further increasing their complexity, which is also reflected in microalgal polyamine metabolic genes [40].

Putrescine (Put), spermidine (Spd), and Spermine (Spm) are typically found in most cells as common polyamines. In some plants and microorganisms, uncommon polyamines such as diaminopropane (Dap), cadaverine (Cad), nor-spermidine (NSpd), thermospermine (TSpm), branched chain polyamines, and long-chain polyamines (LCPAs) have also been identified (Figure 1) [41,42,43,44,45,46,47,48,49,50,51,52,53,54,55,56,57].

Polyamines exist in different forms in cells. This may be in a free form, an insoluble form bound to structural molecules, or covalently conjugated to proteins and peptides. Therefore, the extraction method may influence the detection of cellular polyamines. 

Free-form polyamines transported between cell organelles are the most easily extracted and analyzed polyamines. Previous reports have suggested that microalgae have more types of free-form polyamines than other organisms [48,49,50,51,52,53,54,55,56,57]. In addition to Put, Spd, and Spm, which are common to animal cells, microalgae also contain Dap, Cad, NSpd, and NSpm, which are usually found in plants and microorganisms (Table 1). NSpd is the most abundant polyamine in many microalgae. The major free-form polyamine in dinoflagellate cells is NSpd/NSpm, which is very different to most organisms where Put and Spd are the dominant free-form polyamines. Although NSpd has been reported to affect the formation of bacterial biofilms and the growth of tumor cells, the physiological function of NSpd in microalgae is still unclear [58,59,60,61].

Burczyk et al. recently analyzed cell wall-conjugated polyamines in *Scenedesmus* and *Chlorella*. These Chlorophyceae have cell walls similar to higher plants, as well as similar conjugated polyamine composition including Put, Spd, and Spm [62]. In contrast, the cell walls of diatoms are not made of cellulose. Their unique cell wall, the frustule, also contains very different polyamines compared to higher plants [44,45,46,47].

Diatoms may be the most important microalgae on earth, contributing more than 40% of marine primary production and more than 20% of CO_2_ fixation globally [31,63]. Since their chloroplasts are derived from a secondary endosymbiotic event, the genome, metabolism, physiology, and structure are very different from organisms in Plantae, which are originated from the primary endosymbiosis. The frustules of diatoms are mainly composed of silicon dioxide (SiO_2_). According to previous studies, polyamines have been extracted from the frustule of *Chaetoceros didymium*, *Coscinodiscus asteromphalus*, *Coscinodiscus granii*, *Cyclotella cryptica, Cylindrotheca fusiformis*, *Nitzschia angularis*, and *Thalassiosira pseudonana* [44,45,46,47]. Even after thousands of years, polyamines could be still measured in the siliceous frustules buried in marine and lake sediments [64,65]. The polyamines found in the frustule are mainly LCPAs [66,67,68,69]. Interestingly, LCPAs are also found in sponge bone needles that are also composed of silicon dioxide [70]. 

LCPAs usually refer to polyamines containing several repeated units attached to Put or Dap (Figure 1). In fact, they have high affinity to siliceous material and promote the precipitation of silica [44,45,46,47]. Several fluorescently-tagged polyamines were synthesized as a staining dye for the study of biomineralization process of the diatom cell wall [67]. Recently, Pawolski et al. has demonstrated that LCPAs are the key component for *in vitro* synthesis silica patterns in diatom-like hierarchically porous plate [47]. Such kind of meso- to microporous inorganic materials derived from natural ingredients may be applied as a biomedical devices or drug carriers with better biocompatibility [71].

Above examples show that the composition and function of microalgae are unique, and the biosynthetic pathways and physiological functions of these distinctive polyamines deserve further study.

## 3. Polyamine Synthetic Pathways and Transport in Microalgae

Few polyamine synthetic enzymes have been well characterized, but the metabolic routes can be predicted from sequenced microalgal genomes. In general, microalgae have complex and unique polyamine synthesis pathways, especially microalgae that produce uncommon polyamines. Since these uncommon polyamines may be closely associated with physiological functions, and may also have potential commercial value for biomedical applications, these pathways merit further study. 

### 3.1. Putrescine Biosynthesis

Putrescine is a diamine and, in most organisms, it is the common precursor for the synthesis of other polyamines. The biosynthesis of Put in most animal cells involves two reactions. First, arginine is converted to ornithine by arginase, then ornithine is further converted to Put by the activity of ornithine decarboxylase (ODC, EC 4.1.1.17). In prokaryotes, arginine is often used to generate agmatine *via* arginine decarboxylase (ADC, EC 4.1.1.19), with Put subsequently being generated from agmatine by two distinct pathways. In the first route, Put is catalyzed in a single-step reaction by agmatinase/agmatine ureohydrolase (AUH, EC 3.5.3.11); in the second route, Put is synthesized via two successive reactions, involving agmatine deiminase/iminohydrolase (AIH, EC 3.5.3.12) and N-carbamoylputrescine amidohydrolase/amidase (NCPAH; EC 3.5. 1.53) (Figure 2).

There are multiple Put anabolic pathways in plant genomes. In addition to the common ODC pathway, ADC/AIH/NCPAH pathways, derived from cyanobacteria, are also found in plants, considered as further proof for the endosymbiotic evolution of plants from cyanobacteria [72].

In microalgae, however, Put biosynthesis is considerably different. Most microalgae possess the ODC pathway, but some also appear to have an incomplete ADC/AIH/NCPAH pathway (Table 2). According to the genome database for the dinoflagellates *Symbiodinium* sp., the ODC pathway is the only Put synthetic route in these species. Homologs of AIH and NCPAH, but not ADC, are present in the genomes of red algae, diatoms, and green algae such as *Bathycoccus prasinos* (Chlorophyceae) and *Thalassiosira pseudonana* (Bacillariophyceae). Moreover, they often have more than two sets of ODC genes (Table 2), hence most microalgae were thought to use ODC as the main Put synthetic pathway. However, it is notable that AUH is also present in the genome of *Phaeodactylum tricornutum* (Bacillariophyceae). Since AUH is derived from bacteria, *P. tricornutum* was proposed to have acquired this gene through horizontal gene transfer. The existence of AUH and the extra ODC genes may increase the levels of Put to meet the unique physiological requirements of microalgae.

Besides bioinformatic approaches, it is also possible to assay enzyme activity directly in microalgal cells, which can also help elucidate the Put synthetic pathways. For example, intracellular ADC activity in *Scenedesmus obliquus* (Chlorophyceae) was found to be extremely low, suggesting that ODC may be the major Put synthetic route in these cells [73]. In contrast, experiments in *Chlorella vulgaris* (Chlorophyceae) showed that ADC activity was approximately four-fold higher than ODC; therefore, in this case, the ADC pathway may be the major Put synthetic route [74]. Indeed, the genus *Chlorella* is the only microalgae to possess a complete ADC/AIH/NCPAH pathway, according to their genomic data (Table 2).

Recently, Tassoni et al. detected ADC enzymatic activity in *Chlamydomonas reinhardtii*, even though the ADC gene is not present in its genome [75]. In general, ODC from the mouse, humans, plants, and some parasites is known to have a very strong preference for ornithine as a substrate [76,77,78]. However, ODC may also have weak enzymatic activity toward Lys and Arg [79,80]. Moreover, there is a unique ODC homolog (PBCV1 DC) present in the genome of the *Paramecium Bursaria* Chlorella virus, which shows a preference for Arg as a substrate. A detailed biochemical study indicated that the Asp-to-Glu substitution in the activity center of PBCV1 DC is responsible for the substrate preference switch [81]. A similar phenomenon is also found in an enzyme isolated from *Bacillus subtilis*, which can use both ornithine and Arg as substrates [82]. This enzyme (NP_389346) belongs to decarboxylase group IV, which also includes ODC and ADC [83]. Since this enzyme displayed both ODC and ADC activity, it was named ornithine/arginine decarboxylase (OAD) [84]. By using bacterial OAD sequences as a template to conduct a comprehensive search of microalgal genomes, we found OAD, but not ADC-like, genes in green algae, red algae, and diatoms. Therefore, we speculate that the substitution of ADC with OAD may be a common phenomenon in microalgae (Table 2). However, further studies are needed to elucidate the true catalytic activity and characteristics of these OADs.

### 3.2. Decarboxylated S-adenosylmethionine Biosynthesis

S-adenosylmethionine (SAM) is an essential metabolite in cells. In addition to serving as a substrate for various methylation reactions, SAM can also be converted into decarboxylated S-adenosylmethionine (dcSAM) by S-adenosylmethionine decarboxylase (SAMDC, EC 4.1.1.50). Subsequently, dcSAM provides its aminopropyl moiety for the synthesis of higher polyamines by a group of enzymes known as aminopropyltransferases (APT). Therefore, the enzymes SAMDC and APT are both important for the biosynthesis of higher polyamines.

In plants and microalgae, SAMDC expression levels influence not only the production of higher polyamines, but also have considerable influence on their physiology, especially the capacity to tolerate stress [85,86,87,88]. Nishibori et al. added methylglyoxal bis-guanylhydrazone (MGBG), an inhibitor of SAMDC, to a red algae culture media and found reduced cellular Spd content, as well as reduced rates of cell growth [50,51]. Frigeri et al. also found that polyamine synthesis genes like SAMDC and ODC are highly expressed at the G1 and G2/M phases in diatom cells [89], all indicating that SAMDC is a key regulatory enzyme in microalgal polyamine synthesis.

In diatoms, a distinctive type of APT may also exhibit SAMDC activity [90]. These proteins contain both APT and SAMDC domains, and it is believed that they could use SAM directly as a substrate to synthesize higher polyamines. There are multiple homologs of SAMDC-APT fusion genes in the genome of a single diatom, suggesting that these distinctive genes may have key roles in the synthesis of special polyamines, like LCPAs, in the diatom. However, the biochemical and physiological functions of these genes remain to be characterized.

### 3.3. Spermidine and Spermine Biosynthesis

Apart from SAMDC-APT, most APTs are standalone enzymes which transfer the aminopropyl group from dcSAM to polyamines. Each APT has its own substrate preference. For example, human spermidine synthase (HsSDS) and spermine synthase (HsSMS) specifically use Put and Spd as aminopropyl group acceptors, respectively [78,91]. In contrast, some APTs are promiscuous in substrate selectivity, such as the spermidine synthase from the bacterium *Thermotoga maritima* (TmSDS). TmSDS can also transfer aminopropyl groups to other polyamines such as NSpd, Spd, Cad, and Dap [92]. Based on analysis of protein structure, the gate keeper sequence of the substrate binding pocket (D173–F185) in HsSDS is very important for substrate selectivity [75]. Compared to HsSDS, a Pro has been replaced with an Ala at the gate keeper sequence of TmSDS, a substitution that is considered to increase the flexibility of substrate binding [93].

According to genomic information, most microalgae have multiple APT genes, consistent with the variety of polyamines usually produced by microalgae (Table 1). Although the activity of some enzymes can be predicted (Table 2), their substrate preferences may change since the gate keeper sequences are very different. Thus, it is necessary to further characterize the properties of these APT enzymes. To date, only one APT from *Thalassiosira pseudonana* (XP_002294888) has been characterized and had its activity as a spermidine synthase confirmed [94].

### 3.4. Thermospermine Biosynthesis

The molecular weights of TSpm and Spm are identical, but their linked aminopropyl groups have a different configuration. TSpm, an uncommon polyamine, was first discovered in the thermophilic bacteria *Thermus thermophilus* [95], and later found in other bacteria and plants [41,96]. The amount of Tspm in cells is generally less than 5% of the total polyamines, but a few thermophilic bacteria such as *Thermus brockianus* can reach about 41% [41]. There is still no conclusion about the physiological function of TSpm in thermophilic bacteria [97,98]. However, TSpm plays a prominent role in plant physiology. It is related to the elongation of stems and the differentiation of xylem in *Arabidopsis thaliana*, although the content of Tspm in cells is not as much as that of Spm [99,100].

TSpm is synthesized by thermospermine synthase (TSMS, EC 2.5.1.79), which also belongs to the APT protein family [90]. TSMS is very similar to SMS both in primary sequence and biosynthetic mechanism; both use Spd as an acceptor substrate for the dcSAM aminopropyl group, but the products are different. A possible explanation comes from a recent study, which solved the crystal structure of TSMS from *Medicago truncatula* (MtTSMS) [101]. First, the dcSAM binding site may be important for the differential enzymatic activity. There are three highly-conserved amino acids in TSMS, namely Gln85, Asp109, and Glu129, which are responsible for binding to the terminal amine of the dcSAM aminopropyl group and ribose. In SMS, these residues are His85, Glu109, and Asp129. Although these variants do not change the binding mode of dsSAM, they affect the binding orientation of polyamines by slightly sifting the binding position. In addition, TSMS has a Trp255 in the middle of the polyamine binding pocket, while most SMS have Ile at the same position. The hydrophobic Trp255 pushes the middle amine group closer to the catalytic Asp178 and, in this way, the aminopropyl and aminobutyl moieties of Spd are well orientated to generate TSpm instead of Spm [101].

Moreover, a distinctive type of TSMS which is promiscuous in substrate selectivity has been identified in thermophilic archaea [102]. It not only converts Spd to TSpm, but also generates Nspd from Dap, even adding additional aminopropyl groups to produce longer polyamines. Under optimized conditions, this type of TSMS can synthesize caldoheptamine, an LCPA containing seven aminopropyl groups.

Previous studies have suggested that TSpm may be an uncommon polyamine in microalgae (Table 1). For example, Hamana et al. analyzed 18 green algal species, but only trace amounts of TSpm could be detected in 3 species [57]. Therefore, TSpm may be a less important polyamine in microalgae. However, bioinformatics analysis showed that TSMS homologous genes from different evolutionary origins are widely present in the genomes of various microalgae (Table 2). Phylogenetic analysis indicated that TSMS homologs of green and red algae are more like higher plants, while those of diatoms are closer to archaea [90]. The expression profiles, enzymatic activity, and physiological functions of these TSMS homologs are still not fully understood. The low expression of TSMS observed in microalgae may result from a lack of knowledge of the induction condition of this gene. Indeed, Knott et al. detected TSMS enzymatic activity in *Thalassiosira pseudonana* [103], and this enzyme was further examined and identified as a TSMS [94]. Nevertheless, additional TSMS homologs must be characterize to reveal the physiological roles of TSpm in different microalgae.

### 3.5. Norspermidine and Diaminopropane Biosynthesis

Although NSpd is widely present in most microalgae (Table 1), the corresponding synthetic pathway remains unclear. According to a previous study of *Vibrio cholera*, Nspd can be synthesized from Dap and aspartate β-semialdehyde (ASA) [59]. Dap is first linked with ASA by carboxynorspermidine dehydrogenase (CANSDH) to form carboxynorspermidine (CANS). This product is then converted to NSpd by carboxynorspermidine decarboxylase (CANSDC). However, these enzymes are not present in microalgae, indicating that other NSpd synthetic routes may be active.

In plants, Dap can be converted from Spd and Spm by polyamine oxidase (PAO). Four types of PAO have been identified in *Arabidopsis thaliana*. Among them, AtPAO1, 2, and 4 can generate Dap after polyamine oxidation [104,105,106,107,108]. Several putative PAO genes have been identified (Table 2) using AtPAO sequences as templates for a BLAST search of microalgal genomes, suggesting these PAO homologs in microalgae may produce Dap as a precursor of NSpd. Furthermore, Green et al. overexpressed a fused SAMDC-APT gene from marine bacteria in *Escherichia coli*, which resulted in the production of NSpd in *E*. *coli* cells [109]. As mentioned above, these types of SAMDC-APT fusion enzymes are also found in marine diatoms [90], and further studies are needed to determine whether SAMDC-APT fusion enzymes have key roles in NSpd synthesis in diatoms.

### 3.6. Transport of Microalgal Polyamines 

In addition to the regulation of polyamine biosynthetic enzyme activity, manipulating various cell membrane- or organelle-localized polyamine transporters is also important to maintain cell polyamine homeostasis. At least six types of polyamine transporters have been identified in *E. coli*, including two Put uptake transporters (PotFGHI system, PuuP), one Spd uptake transporter (PotABCD system), two antiporters (PotE, CadB), and one Spd secretion protein (MdtJI system) [29]. In *Saccharomyces cerevisiae*, the polyamine transport system consists of four polyamine uptake proteins (DUR3, SAM3, GAP1, and AGP2) and five polyamine secretion proteins (TPO1–5) [29]. In *A. thaliana*, there are three polyamine uptake transporters (AtPUT1–3) located in the endoplasmic reticulum, Golgi apparatus, and plasma membrane [110,111].

Notably, AtPUT homologs are present in almost all microalgal genomes (Table 2). Many microalgae can absorb and utilize polyamines from the environment, although the putative polyamine transporters have not been extensively studied [50,51,75,112,113,114]. Liu et al. recently reported that *Thalassiosira pseudonana* can also secrete a variety of polyamines extracellularly [52], suggesting additional polyamine transporter may also exist in diatoms. Since diatoms are the dominant phytoplankton in the ocean [115], polyamines secreted by diatoms may have some influence on the marine ecosystem. 

Nitrogen is often the most limiting nutrient for microalgal growth in the oceans [116]. Therefore, the availability of dissolved organic nitrogen (DON), which can be used directly by phytoplankton, has a great impact on the primary productivity of marine ecosystems [52,75,115]. Indeed, polyamines are one of the most important sources of DON in the oceans [117]. According to a study by Liu et al., Put, Spd, and Spm are the most abundant polyamines in the marine environment, while other polyamines (i.e., Dap, Cad, NSpd, and NSpm) are only detected occasionally [118]. It is also known that local changes in polyamine concentration and content are usually related to algal blooms or the appearance of specific marine bacteria [117,118,119,120,121]. For example, the environmental polyamine concentration increased during an algal bloom dominated by *Skeletonema costatum* (Bacillariophyceae) [122]. Specifically, the concentration of Spd increased during a *Chattonella antiqua* (Raphidophyceae) red tide, in which Spd was the most abundant cellular polyamine detected [81,82]. Notably, NSpm, which is relatively rare in the environment, was detected during a *Gymnodinium mikimotoii* (Dinophyceae) algal bloom [53,117].

Adding Put and Spd to culture medium containing a mixture of plankton collected from the natural environment was reported to significantly increase the biomass of the blue-green bacterium *Mycrocystis aeruginosa* [123]. It was also reported that the population of Roseobacter, a type of marine bacteria, increased concomitantly with a rise in seawater polyamine concentration [114]. Moreover, environmental bacteria, like *Pseudomonas* sp., can also inhibit the growth of *Microcystis flosaquae* by secreting Cad [124]. These examples strongly suggest that polyamines are not only used as DON in the environment, but also affect the concentrations of specific organisms in aquatic ecosystems. A similar phenomenon was reported by Amin et al., who revealed the existence of a mutualistic relationship between diatoms and certain *Roseobacter* species. The diatom secretes organosulfur, which can be absorbed by a specific type of Roseobacter. In a feedback mechanism, the Roseobacter then releases some plant hormones that promote the growth of the diatoms [125]. Since polyamines can influence the growth of specific types of microalgae and marine bacteria, they may also have roles as informational chemicals (infochemicals) in ecosystems.

## 4. The Physiological Roles of Polyamines in Microalgae

### 4.1. Promotion of Cell Division and Growth

In green algae, the relationship between polyamines and the cell cycle was first reported half a century ago [126]. The cellular concentration of Put and Spd increased in the S phase, implying that polyamines are closely linked with DNA replication and cell division [54,127,128,129]. A similar phenomenon has been observed in other microalgae, although the types of polyamines involved are different. In *Alexandrium minutum*, cellular levels of Put and Cad increased significantly during the exponential growth phase, while in *Thalassiosira pseudonana* the same was observed with NSpd [52,113]. As with *Chattonella antiqua* and *Heterosigma akashiwo*, Spd was found to be highly correlated with rapid cell growth [50,51]. 

Meanwhile, the addition of polyamines to the culture medium was shown to promote the growth of various microalgae [75,112,113,114]. Notably, the growth of the shellfish poison-producing alga *A*. *minutum* is stimulated by polyamines; however, this is not related to accumulation of algal toxicity, and the addition of Spm may even reduce the content of algal poisons [113]. In contrast, the presence of polyamine synthetic enzyme inhibitors affected considerably the growth of microalgae [50,51,73,74,75,91].

In diatoms, the synthesis of the silica cell wall (frustule) is a crucial step in cell proliferation. Most diatom species will stop growing, or even die, if their frustule cannot be synthesized [130]. Polyamines are known to also be involved in the synthesis of diatom frustule. The addition of ODC inhibitors to *T*. *pseudonana* induced a significant distortion in frustule morphology due to the lack of silicification [91]. A group of distinctive LCPAs that usually consist of a Put with an additional 7–15 aminopropyl groups, can be extracted from the frustule of many diatom species [44,45,46,47]. 

LCPAs may also be covalently conjugated to proteins as a post-translational modification. Silaffin, a protein involved in frustule formation, was found to have better biomineralization capacity after LCPA conjugation [131]. A recently study also indicated that LCPA are essential for the porous silica patterns of frustule [47]. In addition, LCPAs may also be involved in the maintenance of cell membrane configuration in diatoms [132]. Together, this information indicates the importance of LCPAs to marine diatoms. 

### 4.2. Aid to Photosynthesis

The relationship between polyamines and photosynthesis was first revealed in plants [133,134]. In *Avena sativa*, *Zea mays* and other plants, polyamines can be found in various photosynthetic apparatus of chloroplasts, and different types of polyamine have unique distribution patterns [135]. For example, Put was specifically located at the light-harvesting complexes of spinach, while Spm was the major polyamine in photosynthetic system II (PSII) [134]. 

The role of polyamines in chloroplasts is not fully understood. Exogenously applied polyamines can rapidly enter intact chloroplasts and prevent chlorophyll loss during leaf senescence [136]. It was found that polyamines can be covalently conjugated to chlorophyll-bound proteins by plastidial transglutaminase, indicating that this may be one of the mechanisms involved in polyamine modulation of chlorophyll stability [137,138].

Recent studies suggested that polyamines, especially Put, may also facilitate the transportation of CO_2_ [139,140]. In addition, exogenous application of polyamines may also affect the activity of Ribulose-l,5-bisphosphate carboxylase (RuBPC), the key enzyme in the dark reaction, and help to adjust the photosynthetic rate during stress conditions like drought, salt stress, or hypoxia [141,142].

Studies of polyamines and photosynthesis in microalgae have focused mainly on the unicellular green alga, *Scenedesmus obliquus*. Several polyamines were found to be associated with light-harvesting complex II (LHCII) and chlorophyll proteins. The negatively charged LHCII became randomly distributed in the thylakoid membrane and lost their connection with the photosystem II center if there were no polyamines associated [73,143]. Beigbeder et al. also confirmed that consumption of Put is crucial for chlorophyll synthesis [112]. A subsequent study found that a low dose of Spm could also promote the synthesis of chlorophyll, while addition of a high concentration of exogenous Spm could lead to a structural change in the thylakoid membrane system and reduce the amount of chlorophyll [73]. Ioannidis et al. suggested that high amounts of Spm conjugated to LHCII may increase a nonphotochemical quenching effect, resulting in energy dissipation [144]. Therefore, polyamines play a pivotal role in photosynthesis, and different types of polyamine may have different effects.

### 4.3. Relationship with Environmental Stress

Numerous studies in plants and macroalgae have shown that polyamines play important roles in protecting against abiotic stresses [10,145,146,147,148,149]. Like many autotroph organisms, microalgae must to cope with environmental stress in nature, such as a lack of nutrients, temperature fluctuations, and excessive light. According to several independent studies, polyamines might also be important for adaptation to environmental stresses in different types of microalgae [2,52,53,54,55,56,150,151,152,153]. 

In studies of *Scenedesmus obliquus*, UV irradiation was found to result in a decrease in thylakoid-associated Put levels, while the levels of Spm increased. Switching polyamine types in the thylakoid led to an increase in the size of LHCII and amplifying the UV-B effects, thus reduce both photosynthetic efficiency and cell growth [150]. However, addition of exogenous Put inhibited the LHCII size increase and rescued the effects of UV-B, whereas exogenous Spm enhanced UV-B injury. Comparison of an *S*. *obliquus* chlorophyll b-less mutant strain with wild type showed that polyamine level change is a more sensitive marker than xanthophyll when algae are exposed to UV-B irradiation [150]. In fact, light stress led to two-fold increase in Spm levels. It is also found that the attachment of Spm to thylakoids may be important to maintain their structure or function [150].

When the green algae *S*. *obliquus* encounters high salt stress, the size of the antenna complex also increases, and the density of active photosystem II reaction centers is reduced, like that observed under UV-B stress [151]. Exogenous Put may rescue the size of LHCII as well as photosynthetic efficiency [150,151], allowing cell growth to continue under salt stress. In addition to an increase in polyamine concentrations in *Dunaliella salina* under high salt conditions, the simultaneous increase in the expression of plastidial transglutaminase may increase the chlorophyll stability [152]. A similar response was observed in algae growing in an acidic environment [2], showing that changes in the proportion of polyamines in photosynthetic cells can protect microalgae from different environmental stresses. 

Under stress conditions, the concentrations and ratios of polyamines fluctuate not only in the chloroplast, but also in the cytosol, or even in the extracellular environment. In red algae and diatoms, high temperature stress was found to result in an increase in cellular polyamine content, a change in the proportion of polyamines, and the excretion of polyamines [52,55,56]. The changes in polyamine synthesis are also reflected at the level of gene expression. For example, spermidine synthase of *D*. *salina* was found to be highly expressed under high salt stress [153]. 

Regardless of the polyamine synthetic pathway used by microalgae, polyamine synthesis requires energy, as well as other important nutrients like amino acids, making nutrition limitation stress a major factor affecting polyamine content in microalgae. Nitrogen is one of the main elements required for polyamine synthesis, and total cellular polyamine content was seen to decrease under nitrogen-limited conditions [154]. The polyamine content was also decreased by a lack of other nutrient with only exception being reported by Liu et. al., who observed a slight increase in polyamine levels in diatoms cultured in the absence of silicate [52].

In addition to the synthesis of polyamines for their own requirements, some microalgae also secrete specific polyamines into the surrounding environment. For example, *Thalassiosira pseudonana* secretes NSpd constitutively under normal culture conditions [52]. Under nitrogen depletion, *T*. *pseudonana* still secreted polyamine despite a rapid decrease in the concentration of cellular polyamines, but Spd was secreted instead of NSpd [52]. These data suggest that polyamines may not only be regulatory molecules in individual microalgal cells, but may also play roles as extracellular chemical messengers.

## 5. Conclusions and Future Prospects

Most studies on microalgae have focused on ecological impacts. Recently, however, research has focused more on the biochemistry and physiology of microalgae due to the commercial potential of the microalgal industry. Polyamines are essential components for the growth of microalgae and they play important roles in the adaptation to various environmental stresses. Therefore, more research should be geared towards investigating the synthetic pathways and physiological function of polyamines in microalgae.

Due to the multiple evolutionary origins of microalgae, their polyamine contents and metabolic pathways are more complex than other organisms. Nevertheless, polyamines are involved in the cell cycle, photosynthesis, and stress adaptation in both microalgae and higher plants. However, microalgae also produce distinctive polyamines with roles in distinct physiological functions. For example, the LCPAs of diatoms are associated with the construction of a unique silica cell wall [44,45,46,47]. Additional comprehensive studies are needed to reveal the functions of different microalgal polyamines.

First, additional polyamine metabolism-related enzymes in microalgae should be characterized, since relatively few microalgal genes involved in polyamine synthesis have been identified to date. To clarify the enzymes responsible for the synthesis of a specific polyamine, the expression profiles of these genes could be monitored in microalgae under specific physiological conditions. 

In recent years, an increasing number of studies have observed a correlation between the environmental polyamine content and algal blooms [52,118]. To further understand how environmental polyamines are secreted or used by microalgae, genes related to polyamine transport, storage, and conjugation must be studied extensively. Since specific polyamines may have roles as nutrients or signaling molecules that affect specific organisms in the ecosystem, the expression profile of these genes may be used as an indicator to monitor and predict population changes in the environment. 

Finally, we also noticed that microalgae can produce many uncommon polyamines. Numerous studies have suggested that these uncommon polyamines may have the ability to kill cancer cells [30,155,156]. In addition to the potential medical applications, some LCPAs can promote the formation of silica nanomaterials, which also has potential applications [44,45,46,47]. In the future, there may be commercial value in cultivating microalgae to generate uncommon polyamines. The microalgal enzymes, or even entire metabolic pathways, involved in the synthesis of distinctive polyamines may be transferred to a target organism for mass production via a synthetic biology approach. Therefore, the study of polyamines in microalgae is not only beneficial for microalgal and marine ecology research, but also useful for biotechnology and industry.

## Figures and Tables

**Figure 1 marinedrugs-17-00001-f001:**
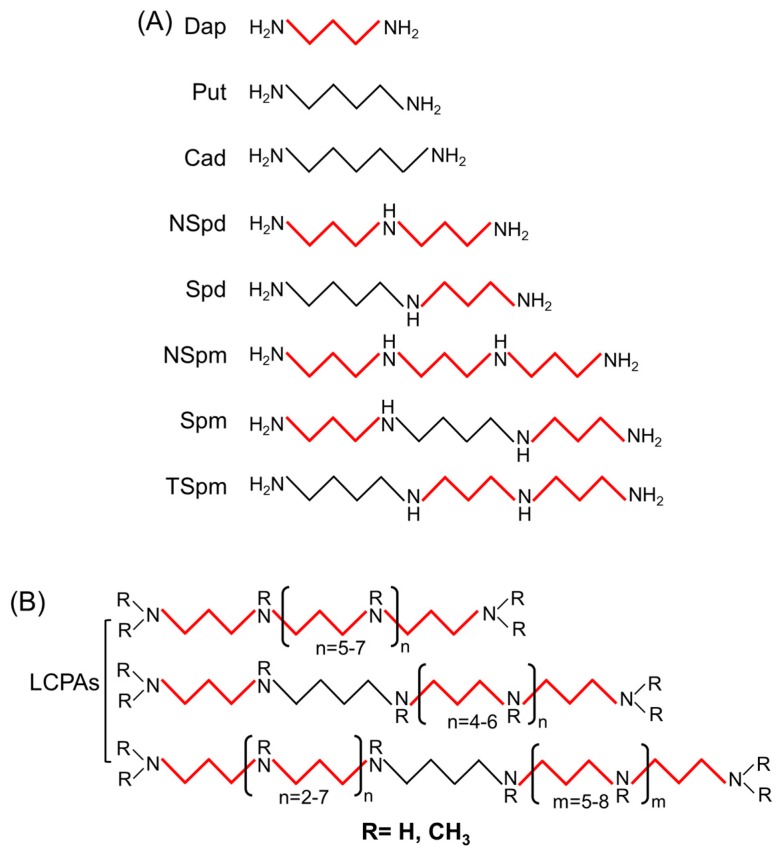
Molecular structure of various microalgal polyamines. (**A**) The polyamines commonly found in various microalgae; (**B**) Special long-chain polyamines present in diatoms. The thick red line in the chemical structure represents the aminopropyl groups while the fine black line represents the aminobutyl or aminopentyl groups. Dap, diaminopropane; Put, putrescine; Cad, cadaverine; NSpd, nor-spermidine; Spd, spermidine; NSpm, nor-spermine; Spm, spermine; TSpm, thermospermine; LCPAs, long-chain polyamines.

**Figure 2 marinedrugs-17-00001-f002:**
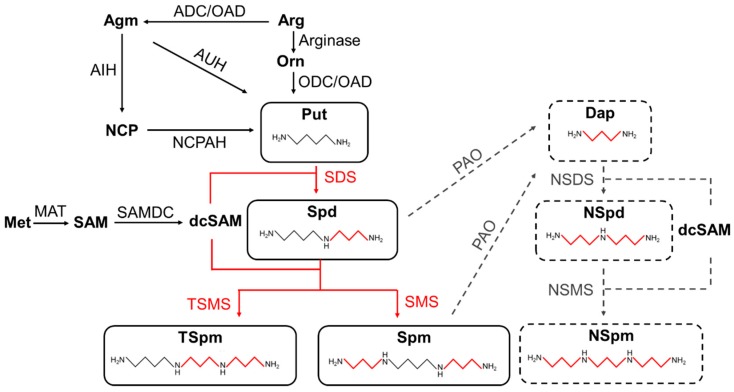
Predictive pathway for polyamine synthesis in microalgal cells. The richness and diversity of polyamines in microalgal cells may be derived from metabolic pathways comprising the following polyamine synthase enzymes. Metabolic pathways that are still hypothetically staged are indicated by dashed lines. Among them, the anabolic pathways of the long-chain polyamines derived from diatoms are still unclear. Agm, agmatine; Arg, arginine; Met, methionine; Orn, ornithine; NCP, N-carbamoylputrescine; SAM, S-adenosylmethionine; dcSAM, decarboxylated S-adenosylmethionine; Dap, diaminopropane; Put, putrescine; NSpd, nor-spermidine; Spd, spermidine; NSpm, nor-spermine; Spm, spermine; TSpm, thermospermine; ADC, arginine decarboxylase; AIH, agmatine iminohydrolase; AUH, agmatine ureohydrolase; MAT, methionine adenosyl transferase; NACPH, N-carbamoylputrescine amidohydrolase; NSDS, nor-spermidine synthase; NSMS, nor-spermine synthase; ODC, ornithine decarboxylase; OAD, ornithine/arginine decarboxylase; PAO, polyamine oxidase; SAMDC, S-adenosylmethionine decarboxylase; SDS, spermidine synthase; SMS, spermine synthase.

**Table 1 marinedrugs-17-00001-t001:** Composition of cellular polyamines in microalgae.

			Polyamines							
Class	Species	Strain	Dap	Put	Spd	NSpd	NSpm	Cad	Spm	Reference
Chlorophyceae	*Chlamydomonas reinhardtii*	IAM C-9	N.D.	91.2%	1.1%	7.6%	N.D.	N.D.	N.D.	[48]
	*Chlamydomonas moewusii*	IAM C-15	N.D.	85.3%	1.0%	1.2%	N.D.	12.4%	N.D.	[48]
	*Chlamydomonas applanata*	IAM C-214	N.D.	87.8%	2.4%	9.8%	N.D.	N.D.	N.D.	[48]
	*Bracteacoccus giganteus*	IAM C-388	N.D.	27.3%	14.5%	54.5%	N.D.	3.6%	N.D.	[48]
	*Chlorococcum echinozygotum*	IAM C-386	N.D.	84.0%	3.4%	12.6%	N.D.	N.D.	N.D.	[48]
	*Chlorogonium elongatum*	IAM C-295	N.D.	50.0%	27.5%	17.5%	N.D.	N.D.	5.0%	[48]
	*Coccomyxa dispar*	IAM C-137	N.D.	3.4%	88.1%	8.5%	N.D.	N.D.	N.D.	[48]
	*Errerella bornhemiensis*	IAM C-581	N.D.	1.5%	32.8%	62.0%	N.D.	3.6%	N.D.	[48]
	*Eudorina illinoisensis*	IAM C-596	N.D.	56.8%	25.0%	15.2%	N.D.	N.D.	3.0%	[48]
	*Gonium pectorale*	IAM C-598	N.D.	13.5%	2.7%	10.8%	N.D.	73.0%	N.D.	[48]
	*Haematococcus lacustris*	IAM C-392	42.8%	10.7%	21.4%	24.1%	N.D.	1.1%	N.D.	[48]
	*Lobomonas piriformis*	IAM C-584	N.D.	37.5%	41.7%	20.8%	N.D.	N.D.	N.D.	[48]
	*Oedogonium obesum*	IAM C-348	N.D.	11.8%	5.9%	14.7%	N.D.	67.6%	N.D.	[48]
	*Dunaliella salina*	IAM C-522	N.D.	15.3%	84.7%	N.D.	N.D.	N.D.	N.D.	[48]
	*Dunaliella bardawil*	ATCC 30861	N.D.	13.3%	82.2%	N.D.	N.D.	4.4%	N.D.	[48]
	*Dunaliella primolecta*	IAM C-525	N.D.	39.3%	60.7%	N.D.	N.D.	N.D.	N.D.	[48]
Chlorophyceae	*Volvox aureus*	IAM C-419	N.D.	80.2%	6.9%	8.8%	N.D.	4.1%	N.D.	[48]
	*Scenedesmus acutus*	IAM C-64	N.D.	67.0%	3.6%	29.4%	N.D.	N.D.	N.D.	[48]
	*Scenedesmus vacuolatus*	IAM C-104	2.4%	34.9%	7.2%	26.5%	N.D.	28.9%	N.D.	[48]
Prasinophyceae	*Mesostigma viride*	NIES-296	N.D.	66.7%	33.3%	N.D.	N.D.	N.D.	N.D.	[48]
	*Monomastix minuta*	NIES-255	N.D.	3.2%	96.8%	N.D.	N.D.	N.D.	N.D.	[48]
	*Nephroselmis olivacea*	NIES-483	N.D.	72.7%	27.3%	N.D.	N.D.	N.D.	N.D.	[48]
	*Nephroselmis viridis*	NIES-486	N.D.	42.3%	57.7%	N.D.	N.D.	N.D.	N.D.	[48]
	*Pyramimonas amylifera*	NIES-251	N.D.	50.0%	50.0%	N.D.	N.D.	N.D.	N.D.	[48]
	*Pyramimonas parkeae*	NIES-254	N.D.	50.0%	50.0%	N.D.	N.D.	N.D.	N.D.	[48]
	*Tetraselmis cordiformis*	NIES-18	N.D.	88.2%	11.8%	N.D.	N.D.	N.D.	N.D.	[48]
Trebouxiophyceae	*Auxenochlorella protothecoides*	ATCC 30407	N.D.	N.D.	71.8%	N.D.	N.D.	N.D.	28.2%	[48]
	*Ankistrodesmus angustus*	IAM C-548	N.D.	48.7%	7.1%	13.3%	N.D.	31.0%	N.D.	[48]
	*Ankistrodesmus falcatus*	IAM C-304	N.D.	23.6%	15.2%	33.8%	0.3%	27.0%	N.D.	[48]
	*Ankistrodesmus nannoselene*	IAM C-305	N.D.	29.0%	29.0%	21.0%	1.6%	11.3%	8.1%	[48]
	*Chlorella kessleri*	IAM C-531	N.D.	68.6%	31.4%	N.D.	N.D.	N.D.	N.D.	[48]
	*Chlorella sorokiniana*	IAM C-212	N.D.	41.0%	53.3%	2.5%	N.D.	3.3%	N.D.	[48]
	*Chlorella vulgaris*	IAM C-27	N.D.	48.9%	51.8%	1.2%	N.D.	2.1%	N.D.	[48]
Trebouxiophyceae	*Watanabea reniformis*	IAM C-211	N.D.	50.0%	50.0%	N.D.	N.D.	N.D.	N.D.	[48]
	*Hydrodictyon reticulatum*	IAM C-335	N.D.	33.3%	55.6%	11.1%	N.D.	N.D.	N.D.	[48]
Chlorarachniophyceae	*Chlorarachnion reptans*	NIES-624	N.D.	32.0%	28.0%	24.0%	4.0%	N.D.	12.0%	[49]
Chrysophyceae	*Poterioochromonas malhamensis*	IAM CS-1	N.D.	64.3%	11.9%	N.D.	N.D.	23.8%	N.D.	[49]
	*Ochromonas danica*	IAM CS-4	N.D.	40.1%	17.7%	N.D.	N.D.	40.5%	0.8%	[49]
	*Ochromonas minuta*	IAM CS-5	N.D.	11.8%	26.5%	N.D.	N.D.	60.1%	1.7%	[49]
Desmophyceae	*Prorocentrum micans*	NIES-12	2.0%	N.D.	N.D.	31.1%	38.9%	27.8%	N.D.	[49]
Euglenoidea	*Euglena gracilis*	IAM E-6	N.D.	20.9%	38.4%	38.4%	2.3%	N.D.	N.D.	[49]
	*Euglena viridis*	IAM E-11	N.D.	49.2%	33.4%	6.8%	2.4%	12.4%	N.D.	[49]
	*Euglena mutabilis*	NIES-286	5.0%	53.2%	25.3%	15.2%	1.2%	N.D.	N.D.	[49]
	*Phacus agilis*	NIES-387	22.9%	48.2%	17.9%	9.2%	1.8%	N.D.	N.D.	[49]
	*Trachelomonas* sp.	Gunma	17.2%	20.1%	20.1%	12.7%	29.9%	N.D.	N.D.	[49]
Eustigmatophyceae	*Nannochloropsis oculata*	IAM ST-4	N.D.	33.0%	67.0%	N.D.	N.D.	N.D.	N.D.	[49]
	*Vischeria punctata*	IAM X-4	1.0%	12.9%	47.3%	6.5%	N.D.	30.8%	1.5%	[49]
	*Vischeria stellata*	IAM X-5	N.D.	23.2%	46.3%	6.8%	N.D.	23.7%	N.D.	[49]
Raphidophyceae	*Chattonella antiqua*	(Hada) Ono	N.A.	2.9%	82.0%	<0.1%	N.A.	N.A.	14.2%	[50]
	*Heterosigma akashiwo*	-	N.A.	2.0%	N.D.	N.D.	N.A.	N.A.	98.0%	[51]
Dinophyceae	*Amphidinium carterae*	NIES-331	<0.1%	N.D.	N.D.	37.7%	51.9%	9.4%	N.D.	[49]
	*Peridinium willei*	NIES-304	1.9%	N.D.	N.D.	74.1%	18.5	5.5%	N.D.	[49]
	*Glenodiniopsis uliginosa*	NIES-463	0.8%	N.D.	N.D.	57.9%	37.2%	4.1%	N.D.	[49]
	*Amphidinium carterae*	CCMP 1314	N.D.	N.D.	N.D.	N.D.	100.0%	N.A.	N.D.	[52]
	*Gymnodinium mikimotoii*	-	N.A.	0.5%	N.D.	0.5%	98.8%	N.D.	N.D.	[53]
	*Alexandrium minutum*	T1	31.3%	N.D.	6.3%	35.4%	N.A.	13.2%	13.9%	[54]
Rhodophyta	*Cyanidium caldarium*	RK-1	N.D.	57.3%	33.2%	N.D.	N.D.	N.D.	9.4%	[55]
Bacillariophyceae	*Nitzschia palea*	IAM B-18	3.1%	82.4%	6.1%	0.8%	N.D.	7.6%	N.D.	[49]
	*Nitzschia closterium*	IAM B-16	N.D.	15.3%	81.6%	3.1%	N.D.	N.D.	N.D.	[49]
	*Phaeodactylum tricornutum*	IAM B-14	N.D.	25.0%	21.9%	37.5%	12.5%	3.1%	N.D.	[49]
	*Thalassiosira pseudonana*	CCMP 1335	8.2%	32.2%	9.2%	43.0%	7.3%	N.A.	N.D.	[52]
	*Chaetoceros* sp.	CCMP 199	N.D.	76.4%	23.6%	N.D.	N.D.	N.A.	N.D.	[52]
	*Skeletonema costatum*	(Grev.) Cleve	N.A.	41.2%	49.6%	N.A.	N.A.	N.A.	8.6%	[56]

N.D. indicates not detectable. N.A. indicates not available.

**Table 2 marinedrugs-17-00001-t002:** Predictive genes related to microalgal polyamine synthesis.

	EC 4.1.1.17	EC 4.1.1.19	EC 3.5.3.12	EC 3.5.1.53		EC 3.5.3.11	EC 3.5.3.1	EC 4.1.1.50	EC 2.5.1.6	EC 2.5.1.79	EC 2.3.1.57	EC 1.5.3.11	
	ODC	ADC	AIH	NCPAH	OAD	AUH	Arginase	SAMDC	SDS	TSMS	SSAT	PAO	PUT
**Diatom**													
*Phaeodactylum tricornutum*	XP_002176410.1 XP_002180545.1	absence	XP_002180809.1	XP_002182987.1	XP_002178296.1	XP_002184908.1	XP_002182650.1	XP_002177188.1	XP_002185179.1 XP_002185737.1	PID 51460	PID13208	XP_002180966.1 XP_002178648.1 XP_002183075.1 XP_002186444.1	XP_002178793.1 XP_002177227.1
*Thalassiosira* *pseudonana*	XP_002287587.1	absence	absence	absence	absence	absence	XP_002296117.1	XP_002290893.1	XP_002294888.1 XP_002287929.1	XP_002294468.1	absence	XP_002290344.1	XP_002296569.1 XP_002290859.1 XP_002292843.1
*Fragilariopsis cylindrus*	OEU12205.1 OEU18440.1	absence	OEU14792.1	OEU22892.1	OEU23567.1	absence	OEU18626.1	OEU18640.1	OEU12592.1 OEU17362.1 OEU21325.1 OEU20819.1	OEU18441.1	OEU14661.1	OEU21990.1 OEU17981.1 OEU15358.1	OEU17653.1
**Green algae**													
*Chlamydomonas reinhardtii*	XP_001697502.1 XP_001698872.1	absence	XP_001700262.1 XP_001689799.1	XP_001692986.1 XP_001690094.1	XP_001695844	absence	absence	XP_001693327.1	XP_001702843.1 ADF43120.1 ADF43160.1	XP_001696651.1	absence	XP_001698304.1 XP_001694551.1 XP_001701014.1	PNW82747.1 XP_001701580.1
*Micromonas pusilla*	XP_003057072.1	XP_003056064.1	absence	absence	absence	absence	absence	XP_003057072.1	XP_003061691.1	XP_003059083.1	absence	XP_003062543.1 XP_003063969.1	absence
*Bathycoccus prasinos*	XP_007511572.1	absence	absence	absence	absence	XP_007512226.1	absence	XP_007514813.1	XP_007509632.1	XP_007513075.1	absence	XP_007508374.1	XP_007511943.1 XP_007514018.1
*Chlorella variabilis*	XP_005847800.1	XP_005845766.1	XP_005851725.1	XP_005852065.1	XP_005845245.1	absence	absence	absence	XP_005845386.1	absence	absence	XP_005845791.1 XP_005844686.1 XP_005851294.1 XP_005843020.1 XP_005850098.1 XP_005847280.1	XP_005842903.1
*Volvox carteri*	XP_002958645.1 XP_002952872.1	absence	XP_002953838.1	XP_002950768.1	XP_002952846.1	absence	absence	XP_002957689.1	XP_002946514.1 ADI46921.1	XP_002954461.1	absence	XP_002957050.1 XP_002953252.1 XP_002954284.1 XP_002954733.1	XP_002953866.1
**Red algae**													
*Cyanidioschyzon merolae*	XP_005535051.1	absence	XP_005536903.1	XP_005537292.1	XP_005536378.1	absence	absence	XP_005537448.1	XP_005537953.1 XP_005538515.1	absence	absence	XP_005537515.1	XP_005538369.1
*Galdieria sulphuraria*	XP_005706808.1 XP_005703430.1	absence	XP_005703407.1 XP_005702449.1 XP_005709153.1 XP_005706917.1	XP_005705431.1 XP_005702451.1 XP_005702268.1 XP_005702450.1	XP_005708583.1	absence	absence	XP_005709363.1	XP_005702905.1 XP_005709061.1 XP_005708678.1	absence	XP_005706745.1	XP_005706126.1 XP_005708277.1	XP_005705642.1 XP_005706720.1
**Dinoflagellate**													
*Symbiodinium* sp.	OLQ02213.1	absence	OLQ00208.1 OLP82902.1 OLP82905.1	absence	absence	absence	OLP95716.1	OLP91558.1	OLQ06968.1 OLQ06304.1 OLQ09646.1 OLP89525.1 OLQ12296.1	OLQ10948.1	absence	OLQ02365.1 OLP81913.1 OLQ06107.1 OLP88294.1	OLP95266.1 OLP96520.1 OLP96519.1

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
