# Peer review of "Polyamines in Microalgae: Something Borrowed, Something New"

_marinedrugs, 2018, doi:10.3390/md17010001_

Round 1
Reviewer 1 Report
In this article, authors present a review about polyamines in microalgae. This review summarizes research on the polyamines found in microalgae, their composition, transport and pathways, and their physiological roles. Future prospect is also discussed.
Many references about polyamines can be found in bibliography. Specifically, this revision is similar to the one published in 2015 in J. Phycol. 51, 838–849: “Polyamines in Macroalge: Advanced and future perspectives”, but in this case is focused on polyamines of microalgae and could be considered as a complementary review.
The paper is well written and understandable and in my opinion, the manuscript has the quality standard required to be published in Marine Drugs Journal. I propose this manuscript to be accepted in present form.
Author Response
Common point
In this article, authors present a review about polyamines in microalgae. This review summarizes research on the polyamines found in microalgae, their composition, transport and pathways, and their physiological roles. Future prospect is also discussed.
Many references about polyamines can be found in bibliography. Specifically, this revision is similar to the one published in 2015 in J. Phycol. 51, 838–849: “Polyamines in Macroalge: Advanced and future perspectives”, but in this case is focused on polyamines of microalgae and could be considered as a complementary review.
The paper is well written and understandable and in my opinion, the manuscript has the quality standard required to be published in Marine Drugs Journal. I propose this manuscript to be accepted in present form.
Response: Thank you very much for your opinion.
Reviewer 2 Report
The Manuscript provide a sound review on polyamine metabolism in microalgae.
The Manuscript contain significant contributions to the field, and it is well organized and comprehensively described.
Some recent references sould be included and properly discussed:
- Pawolski, D., Heintze, C., Mey, I., Steinem, C., and Kroger, N. (2018). Reconstituting the formation of hierarchically porous silica patterns using diatom biomolecules. J Struct Biol 204, 64-74.
- Mogor, A.F., Ordog, V., Lima, G.P.P., Molnar, Z., and Mogor, G. (2018). Biostimulant properties of cyanobacterial hydrolysate related to polyamines. J Appl Phycol 30, 453-460.
- Lin, B., Ahmed, F., Du, H.M., Li, Z., Yan, Y.C., Huang, Y.H., Cui, M., Yin, Y.H., Li, B., Wang, M.M., et al. (2018). Plant growth regulators promote lipid and carotenoid accumulation in Chlorella vulgaris. J Appl Phycol 30, 1549-1561.
- Annenkov, V.V., Verkhozina, O.N., Zelinskiy, S.N., Shishlyannikova, T.A., Bridoux, M.C., and Danilovtseva, E.N. (2018). Unusual Polyamines from Baikalian Diatoms. Chemistry Select 3, 9708-9713.
- Annenkov, V.V., Danilovtseva, E.N., Pal'shin, V.A., Verkhozina, O.N., Shishlyannikova, T.A., Hickman, G.J., and Perry, C.C. (2018). Fluorescently-tagged polyamines for the staining of siliceous materials. Plant Physiol Biochem 125, 205-211.
- Kumar, M., Kuzhiumparambil, U., Ralph, P.J., and Contreras-Porcia, L. (2017). Polyamines: Stress Metabolite in Marine Macrophytes (Amsterdam: Elsevier Science Bv).
- Jantaro, S., and Kanwal, S. (2017). Low-Molecular-Weight Nitrogenous Compounds (GABA and Polyamines) in Blue-Green Algae (Amsterdam: Elsevier Science Bv).
- Du, H.M., Ahmed, F., Lin, B., Li, Z., Huang, Y.H., Sun, G., Ding, H., Wang, C., Meng, C.X., and Gao, Z.Q. (2017). The Effects of Plant Growth Regulators on Cell Growth, Protein, Carotenoid, PUFAs and Lipid Production of Chlorella pyrenoidosa ZF Strain. Energies 10, 23.
- Annenkov, V.V., Danilovtseva, E.N., Khutsishvili, S.S., Pal'shin, V.A., Polienko, Y.F., Saraev, V.V., Vakul'skaya, T.I., Zelinskiy, S.N., and Grigor'ev, I.A. (2017). Polyamine-based spin probes for the study of siliceous structures. Microporous Mesoporous Mat 242, 74-81.
- Stonik, V., and Stonik, I. (2015). Low-Molecular-Weight Metabolites from Diatoms: Structures, Biological Roles and Biosynthesis. Mar Drugs 13, 3672-3709.
- Sabia, A., Baldisserotto, C., Biondi, S., Marchesini, R., Tedeschi, P., Maietti, A., Giovanardi, M., Ferroni, L., and Pancaldi, S. (2015). Re-cultivation of Neochloris oleoabundans in exhausted autotrophic and mixotrophic media: the potential role of polyamines and free fatty acids. Appl Microbiol Biotechnol 99, 10597-10609.
- Du, J.J., Cheng, S.J., Shao, C., Lv, Y.N., Pu, G.Z., Ma, X., Jia, Y., and Tian, X.J. (2014). Dual Roles of Cadaverine-Producing Pseudomonas sp on Microcystis spp. in Hyper-Eutrophic Water. Curr Microbiol 69, 25-31.
Best regards.
Author Response
Major point
Some recent references sould be included and properly discussed:
1.- Pawolski, D., Heintze, C., Mey, I., Steinem, C., and Kroger, N. (2018). Reconstituting the formation of hierarchically porous silica patterns using diatom biomolecules. J Struct Biol 204, 64-74.
2.- Mogor, A.F., Ordog, V., Lima, G.P.P., Molnar, Z., and Mogor, G. (2018). Biostimulant properties of cyanobacterial hydrolysate related to polyamines. J Appl Phycol 30, 453-460.
3.- Lin, B., Ahmed, F., Du, H.M., Li, Z., Yan, Y.C., Huang, Y.H., Cui, M., Yin, Y.H., Li, B., Wang, M.M., et al. (2018). Plant growth regulators promote lipid and carotenoid accumulation in Chlorella vulgaris. J Appl Phycol 30, 1549-1561.
4.- Annenkov, V.V., Verkhozina, O.N., Zelinskiy, S.N., Shishlyannikova, T.A., Bridoux, M.C., and Danilovtseva, E.N. (2018). Unusual Polyamines from Baikalian Diatoms. Chemistry Select 3, 9708-9713.
5.- Annenkov, V.V., Danilovtseva, E.N., Pal'shin, V.A., Verkhozina, O.N., Shishlyannikova, T.A., Hickman, G.J., and Perry, C.C. (2018). Fluorescently-tagged polyamines for the staining of siliceous materials. Plant Physiol Biochem 125, 205-211.
6.- Kumar, M., Kuzhiumparambil, U., Ralph, P.J., and Contreras-Porcia, L. (2017). Polyamines: Stress Metabolite in Marine Macrophytes (Amsterdam: Elsevier Science Bv).
7.- Jantaro, S., and Kanwal, S. (2017). Low-Molecular-Weight Nitrogenous Compounds (GABA and Polyamines) in Blue-Green Algae (Amsterdam: Elsevier Science Bv).
8.- Du, H.M., Ahmed, F., Lin, B., Li, Z., Huang, Y.H., Sun, G., Ding, H., Wang, C., Meng, C.X., and Gao, Z.Q. (2017). The Effects of Plant Growth Regulators on Cell Growth, Protein, Carotenoid, PUFAs and Lipid Production of Chlorella pyrenoidosa ZF Strain. Energies 10, 23.
9.- Annenkov, V.V., Danilovtseva, E.N., Khutsishvili, S.S., Pal'shin, V.A., Polienko, Y.F., Saraev, V.V., Vakul'skaya, T.I., Zelinskiy, S.N., and Grigor'ev, I.A. (2017). Polyamine-based spin probes for the study of siliceous structures. Microporous Mesoporous Mat 242, 74-81.
10.- Stonik, V., and Stonik, I. (2015). Low-Molecular-Weight Metabolites from Diatoms: Structures, Biological Roles and Biosynthesis. Mar Drugs 13, 3672-3709.
11.- Sabia, A., Baldisserotto, C., Biondi, S., Marchesini, R., Tedeschi, P., Maietti, A., Giovanardi, M., Ferroni, L., and Pancaldi, S. (2015). Re-cultivation of Neochloris oleoabundans in exhausted autotrophic and mixotrophic media: the potential role of polyamines and free fatty acids. Appl Microbiol Biotechnol 99, 10597-10609.
12.- Du, J.J., Cheng, S.J., Shao, C., Lv, Y.N., Pu, G.Z., Ma, X., Jia, Y., and Tian, X.J. (2014). Dual Roles of Cadaverine-Producing Pseudomonas sp on Microcystis spp. in Hyper-Eutrophic Water. Curr Microbiol 69, 25-31.
Response:
We followed the reviewer’s suggestion. In addition to adding the abovementioned references, we also included more important references listed below to improve our manuscript.
13.-Xie, S.S.; Wu, H.J.; Zang, H.Y.; Wu, L.M.; Zhu, Q.Q. Gao, X.W. Plant growth promotion by spermidine-producing Bacillus subtilis OKB105. Molecular Plant-Microbe Interactions 2014, 27, 655–663; DOI:10.1094/MPMI-01-14-0010-R.
14.-Schweikert, K.; Burritt, D.J. Polyamines in macroalge: advanced and future perspectives. J. Phycol. 2015, 51, 838–849;DOI:10.1111/jpy.12325.
15.-Armbrust, E.V. The life of diatoms in the world’s oceans. Nature, 2009, 459, 185–192; DOI:10.1038/nature08057.
16.-Howarth, R.W. Nutrient limitation of net primary production in marine ecosystems. Annual Review of Ecology and Systematics 1988, 19, 89-110.
17.-Podlešáková, K.; Ugena, L.; Spícha, L.; Doleža, K.; Diego, N.D. Phytohormones and polyamines regulate plant stress responses by altering GABA pathway. New Biotechnology 2019, 48, 53-65; DOI:10.1016/j.nbt.2018.07.003.
18.-Casero, R.A.Jr.; Stewart, T.M.; Pegg, A.E. Polyamine metabolism and cancer: treatments, challenges and opportunities. Nature Reviews Cancer 2018, 18, 681–695; DOI:10.1038/s41568-018-0050-3.
19.-Kiech, S.; Pechlaner, R.; Willeit, P.; Notdurfter, M.; Paulweber, B.; Willeit, K.; Werner, P.; Ruckenstuh, C.; Iglseder, B.; Weger, S.; et al. Higher spermidine intake is linked to lower mortality: a prospective population-based study. The American Journal of Clinical Nutrition 2018, 108, 371-380; DOI:10.1093/ajcn/nqy102.
20.-Eisenberg, T.; Abdellatif, M.; Schroeder, S.; Primessnig, U.; Stekovic, S.; Pendl, T.; Harger, A.; Schipke, J.; Zimmermann, A.; Schmidt A.; et al. Cardioprotection and lifespan extension by the natural polyamine spermidine. Nature Medicine 2016, 22, 1428-1438; DOI:10.1038/nm.4222.
21.-Soda, K. Biological Effects of polyamines on the prevention of aging-associated diseases and on lifespan extension. Food Science and Technology Research 2015, 21, 145-157; DOI:10.3136/fstr.21.145.
22.-Bridoux, M.C.; Ingalls, A.E. Structural identification of long-chain polyamines associated with diatom biosilica in a Southern Ocean sediment core. Geochimica et Cosmochimica Acta 2010, 74, 4044–4057; DOI:10.1016/j.gca.2010.04.010.
23- Delalat, B.; Sheppard, V.C.; Ghaemi, S.R.; Rao, S.; Prestidge, C.A.; McPhee, G.; Rogers, M.L.; Donoghue, J.F.; Pillay, V.; Johns, T.G.; et al. Targeted drug delivery using genetically engineered diatom biosilica. Nature Communications 2015, 6, 8791; DOI:10.1038/ncomms9791.
24- Brzezinski, M.A.; Olsonl, R.J.; Chisholm, S.W. Silicon availability and cell-cycle progression in marine diatoms. Mar. Ecol. Prog. Ser. 1990, 67, 83-96.
25-Raymond, J.; Blankenship, R.E. Horizontal gene transfer in eukaryotic algal evolution. Proc. Natl. Aca. Sci. U.S.A. 2003, 100, 7419–7420; DOI:10.1073/pnas.1533212100.
26-A quantitative GC-MS method for three major polyamines in postmortem brain cortex. Journal of Mass Spectrometry 2009, 44, 1203-1210; DOI:10.1002/jms.1597.
27-Balcerzak, W.; Pokajewicz, K.; Wieczorek, P.P. A useful procedure for detection of polyamines in biological samples as a potential diagnostic tool in cancer diagnosis. Applied Cancer Research 2017, 37, 23; DOI:10.1186/s41241-017-0032-x.
28-Bridoux, M.C.; Annenkov, V.V.; Menzel, H.; Keil, R.G.; Ingalls, A.E. A new liquid chromatography/electrospray ionization mass spectrometry method for the analysis of underivatized aliphatic long-chain polyamines: application to diatom-rich sediments. Rapid Communications in Mass Spectrometry 2011, 25, 877–888; DOI:10.1002/rcm.4931.
29-Magnes, C.; Fauland, A.; Gander, E.; Narath, S.; Ratzer, M.; Eisenberg, T.; Madeo, F.; Pieber, T.; Sinner, F. Polyamines in biological samples: rapid and robust quantification by solid-phase extraction online-coupled to liquid chromatography–tandem mass spectrometry. Journal of Chromatography A 2015, 1331, 41-51; DOI:10.1016/j.chroma.2013.12.061.
These references have been cited in following position in the article.
Number in the review article | Position | Number in the review article | Position | ||
Ref. 1 | 55 | L116, L122, L125, L342, L363, L450, L466 | Ref. 16 | 111 | L312 |
Ref. 2 | 19 | L39, L40 | Ref. 17 | 146 | L396 |
Ref. 3 | 20 | L39 | Ref. 18 | 30 | L44, |
Ref. 4 | 57 | L117 | Ref. 19 | 27 | L42 |
Ref. 5 | 59 | L118, L123 | Ref. 20 | 26 | L42 |
Ref. 6 | 145 | L396 | Ref. 21 | 25 | L42 |
Ref. 7 | 17 | L38 | Ref. 22 | 56 | L117 |
Ref. 8 | 21 | L39 | Ref. 23 | 63 | L126 |
Ref. 9 | 58 | L118 | Ref. 24 | 127 | L355 |
Ref. 10 | 61 | L118 | Ref. 25 | 40 | L76 |
Ref. 11 | 126 | L342 | Ref. 26 | 33 | L55 |
Ref. 12 | 120 | L330 | Ref. 27 | 34 | L55 |
Ref. 13 | 18 | L39, L40 | Ref. 28 | 35 | L57 |
Ref. 14 | 144 | L396 | Ref. 29 | 36 | L57 |
Ref. 15 | 110 | L310 | Ref. 30 |
Reviewer 3 Report
Review of the manuscript " Polyamines in microalgae: something borrowed, something new” by Hung-Yun Lin and Han-Jia Lin.
General comments
The manuscript aims to review the recent research progress and the differences in content and diversity of polyamines in microalgae.
The data summarized in this review are interesting because the study of polyamines in microalgae is useful both for microalgal ecology research and their potential applications in biotechnology/industry. Overall, the authors did an excellent job collecting and presenting the data. In my opinion, the article is well structured, and the focus of the article fits well with the focus of the journal. The manuscript may be accepted after some revisions given below.
Specific comments
L 38-39 “…improve the memory deterioration …” I would change this sentence because a little misleading: it seems that PA enhances the memory deterioration instead of improving the memory.
L55. “Microalgae, also known as phytoplankton” it is not true: microalgae are not only phytoplankton, but also microphytobenthos (for example, in table 1, Amphidinium carterae is not a component of phytoplankton but of microphytobenthos). Please fix it or just remove “also known as phytoplankton”Table 1. Please write “Prorocentrum micans” in italics.
Table 1. Please in “Trachelomonas sp.” do not write “sp.” in italics
L100. “Diatoms may be the most important phytoplankton” phytoplankton is too reducing because several importantant diatoms species are benthic. Please fix it.
L101-103. “Since they are derived from a secondary endosymbiotic event, the genomes, metabolism, physiology, and structure event.” Something is wrong with this sentence It has no sense to me. Please check it and revise it.
L218. Given this is the first time this name is indicated in this section (i.e. 3.3), please do not use abbreviation “T. pseudonana” and write “Thalassiosira pseudonana”.
L258. Given this is the first time this name is indicated in this section, please do not use abbreviation “T. pseudonana” and write “Thalassiosira pseudonana”.
L290. Given this is the first time this name is indicated in this section, please do not use abbreviation “T. pseudonana” and write “Thalassiosira pseudonana”.
L292. “….diatoms are the dominant phytoplankton in the ocean” please indicate a reference and consider the benthos as well.
L294. “Nitrogen is often the most limiting nutrient for microalgal growth in the oceans”. Please indicate a reference.
L295. “…dissolved organic nitrogen (DON), which can be used directly by phytoplankton” Please indicate a reference.
L324. Given this is the first time this name is indicated in this section, please do not use abbreviation “T. pseudonana” and write “Thalassiosira pseudonana”.
L 376. Given this is the first time this name is indicated in this section, please do not use abbreviation “S. obliquus” and write “Scenedesmus obliquus”.
L406. Given this is the first time this name is indicated in this section, please do not use abbreviation “T. pseudonana” and write “Thalassiosira pseudonana”.
Author Response
Major point
1) L 38-39 “…improve the memory deterioration …” I would change this sentence because a little misleading: it seems that PA enhances the memory deterioration instead of improving the memory.Response (1): We followed that suggestion with modification of the sentence as “Supplement of polyamine in diet can mitigate age-related memory impairment and even prolong the life of animals” (L41-43).
2) L55. “Microalgae, also known as phytoplankton” it is not true: microalgae are not only phytoplankton, but also microphytobenthos (for example, in table 1, Amphidinium carterae is not a component of phytoplankton but of microphytobenthos). Please fix it or just remove “also known as phytoplankton”
Response (2): We take the advice in removing the phrase “also known as phytoplankton”.
3) Table 1. Please write “Prorocentrum micans” in italics.
4) Table 1. Please in “Trachelomonas sp.” do not write “sp.” in italics
Response (3-4): Your kindly suggestions were followed!
5) L100. “Diatoms may be the most important phytoplankton” phytoplankton is too reducing because several important diatoms species are benthic. Please fix it.
Response (5): We followed that suggestion with modification of the sentence as “Diatoms may be the most important microalgae”.
6) L101-103. “Since they are derived from a secondary endosymbiotic event, the genomes, metabolism, physiology, and structure event.” Something is wrong with this sentence It has no sense to me. Please check it and revise it.
Response (6): We apologized for the mistake of editing. The whole sentence should be “Since they are derived from a secondary endosymbiotic event, the genome, metabolism, physiology, and structure are very different from organisms of Planae, which are originated from the primary endosymbiosis” (L110-112).
7) L218. Given this is the first time this name is indicated in this section (i.e. 3.3), please do not use abbreviation “T. pseudonana” and write “Thalassiosira pseudonana”.
8) L258. Given this is the first time this name is indicated in this section, please do not use abbreviation “T. pseudonana” and write “Thalassiosira pseudonana”.
9) L290. Given this is the first time this name is indicated in this section, please do not use abbreviation “T. pseudonana” and write “Thalassiosira pseudonana”.
Response (7-9): Your kindly suggestions were followed!
10) L292. “….diatoms are the dominant phytoplankton in the ocean” please indicate a reference and consider the benthos as well.
11) L294. “Nitrogen is often the most limiting nutrient for microalgal growth in the oceans”. Please indicate a reference.
12) L295. “…dissolved organic nitrogen (DON), which can be used directly by phytoplankton” Please indicate a reference.
Response (10-12): Your kindly suggestions were followed! The references corresponding to these sentences are shown below.
L 309-310. “….diatoms are the dominant phytoplankton in the ocean” Armbrust, E.V. The life of diatoms in the world’s oceans. Nature, 2009, 459, 185–192; DOI:10.1038/nature08057.
L312. “Nitrogen is often the most limiting nutrient for microalgal growth in the oceans” Howarth, R.W. Nutrient limitation of net primary production in marine ecosystems. Annual Review of Ecology and Systematics 1988, 19, 89-110.
L312-314. “…dissolved organic nitrogen (DON), which can be used directly by phytoplankton”Tassoni, A.; Awad, N.; Griffiths, G. Effect of ornithine decarboxylase and norspermidine in modulating cell division in the green alga Chlamydomonas reinhardtii. Plant. Physiol. Biochem. 2018, 123, 125-131; DOI:10.1016/j.plaphy.2017.12.014.
Liu, Q.; Nishibori, N.; Imai, I.; Hollibaugh, J.T. Response of polyamine pools in marine phytoplankton to nutrient limitation and variation in temperature and salinity. Marine Ecology Progress Series 2016, 544, 93–105; DOI:10.3354/meps11583.
Nishiboria, N.; Matuyama, Y.; Uchida, T.; Moriyama, T.; Ogita, Y.; Oda, M.; Hirota, H. Spatial and temporal variations in free polyamine distributions in Uranouchi Inlet, Japan. Marine Chemistry 2003, 82, 307-314; DOI:10.1016/S0304-4203(03)00076-8.
13) L324. Given this is the first time this name is indicated in this section, please do not use abbreviation “T. pseudonana” and write “Thalassiosira pseudonana”.14) L 376. Given this is the first time this name is indicated in this section, please do not use abbreviation “S. obliquus” and write “Scenedesmus obliquus”.
15) L406. Given this is the first time this name is indicated in this section, please do not use abbreviation “T. pseudonana” and write “Thalassiosira pseudonana”.
Response (13-15): All your kindly suggestions were followed!
Reviewer 4 Report
This manuscript contains mainly a review focused on the biosynthesis details of polyamines. Although well written, I believe because it contains this emphasis on synthesis, it is short on other aspects.
For example, in line 49, the authors simply mention the «technological progress in PA detection», but offer no mini-review how the detection of polyamines is carried out. Are these simple HPLC methods, or require the more expensive LC-MS/MS systems and more skilled personnel?
Table 1. In the legend, this is the «relative composition» of polyamines. Also, the table header should be repeated across pages, as the table is 3 pages long.
Line 372: «Microalgae, as a phytoplankton, are often exposed to high levels of environmental stress in nature,…» The meaning of this sentence is not clear. What is the problem of being ‘a phytoplankton’? Being unicellular means being ‘more exposed’ than complex multicellular organisms?
Line 397: it is mentioned that spermidine synthase was enhanced during salt stress. Does this metabolite have anything to do with an osmolyte role? A wide array of small molecules can have a role as osmolytes. For example, methylamines, are among common osmolytes. Is this the case with polyamines? This was never discussed along the manuscript.
Author Response
Major point
1) For example, in line 49, the authors simply mention the «technological progress in PA detection», but offer no mini-review how the detection of polyamines is carried out. Are these simple HPLC methods, or require the more expensive LC-MS/MS systems and more skilled personnel?
Response (1): We added a short paragraph to review the current polyamine analysis methods in L53-57. “Common polyamines are usually detected by GC or HPLC after fluorescent derivatization. Recently, LC-MS/MS is also frequently applied in polyamine detection, especially for those uncommon polyamines which are difficult to acquire standard samples. Currently, MS is the most sensitive polyamine detection method. The detection limit can be as low as 0.1 ng/mL.”
2) Table 1. In the legend, this is the «relative composition» of polyamines. Also, the table header should be repeated across pages, as the table is 3 pages long.
Response (2): Your kindly suggestions were followed.
3) Line 372: «Microalgae, as a phytoplankton, are often exposed to high levels of environmental stress in nature,…» The meaning of this sentence is not clear. What is the problem of being ‘a phytoplankton’? Being unicellular means being ‘more exposed’ than complex multicellular organisms?
Response (3): We followed that suggestion with modification of the paragraph in “Microalgae, like many autotroph organisms, also need to deal with environmental stress in nature, such as a lack of nutrients, temperature fluctuations, and excessive light” (L396-398)..
4) Line 397: it is mentioned that spermidine synthase was enhanced during salt stress.
Does this metabolite have anything to do with an osmolyte role? A wide array of small molecules can have a role as osmolytes. For example, methylamines, are among common osmolytes. Is this the case with polyamines? This was never discussed along the manuscript.
Response (4): The previous version of this paragraph was not clear. We rewrote as following “In red algae and diatoms, high temperature stress was found to result in an increase in cellular polyamine content, a change in the proportion of polyamines, and the excretion of polyamines. The changes in polyamine synthesis are also reflected at the level of gene expression. For example, spermidine synthase of D. salina was found to be highly expressed under high salt stress” (L419-L423).